# Histone Modifications and Their Targeting in Lymphoid Malignancies

**DOI:** 10.3390/ijms23010253

**Published:** 2021-12-27

**Authors:** Miranda Fernández-Serrano, René Winkler, Juliana C. Santos, Marguerite-Marie Le Pannérer, Marcus Buschbeck, Gaël Roué

**Affiliations:** 1Lymphoma Translational Group, Josep Carreras Leukaemia Research Institute (IJC), 08916 Badalona, Spain; mfernandez@carrerasresearch.org (M.F.-S.); jcarvalho@carrerasresearch.org (J.C.S.); 2Department of Biochemistry and Molecular Biology, Autonomous University of Barcelona, 08014 Barcelona, Spain; 3Chromatin, Metabolism and Cell Fate Group, Josep Carreras Leukaemia Research Institute (IJC), 08916 Badalona, Spain; rwinkler@carrerasresearch.org (R.W.); mlepannerer@carrerasresearch.org (M.-M.L.P.); 4Program of Personalized and Predictive Medicine of Cancer, Germans Trias i Pujol Research Institute (IGTP), 08916 Badalona, Spain

**Keywords:** non-Hodgkin lymphoma, epigenetics, DNA methylation, HAT, HDAC, EZH2, bromodomain inhibitors, drug combination, clinical testing

## Abstract

In a wide range of lymphoid neoplasms, the process of malignant transformation is associated with somatic mutations in B cells that affect the epigenetic machinery. Consequential alterations in histone modifications contribute to disease-specific changes in the transcriptional program. Affected genes commonly play important roles in cell cycle regulation, apoptosis-inducing signal transduction, and DNA damage response, thus facilitating the emergence of malignant traits that impair immune surveillance and favor the emergence of different B-cell lymphoma subtypes. In the last two decades, the field has made a major effort to develop therapies that target these epigenetic alterations. In this review, we discuss which epigenetic alterations occur in B-cell non-Hodgkin lymphoma. Furthermore, we aim to present in a close to comprehensive manner the current state-of-the-art in the preclinical and clinical development of epigenetic drugs. We focus on therapeutic strategies interfering with histone methylation and acetylation as these are most advanced in being deployed from the bench-to-bedside and have the greatest potential to improve the prognosis of lymphoma patients.

## 1. Introduction

The observation that the epigenetic mechanisms controlling the transcription of a wide range of genes involved in B-cell development are frequently dysregulated in lymphoid neoplasms, has recently centered the efforts of hematological cancer researchers worldwide. These alterations can affect DNA methylation, covalent histone modifications, protein recognition modules, and different chromatin remodelers. Among the latter, epigenetic modifications of histones are versatile marks that are intimately connected to lymphomagenesis. The term B-cell non-Hodgkin lymphoma (B-NHL) encompasses different neoplasms characterized by an abnormal proliferation of lymphoid B cells. Here, we provide an update on current efforts to develop therapeutic strategies to interfere with two major post-translational histone modifications altered in B-NHL: histone acetylation and methylation.

### 1.1. Characteristics of the Main B-NHL Subtypes

B-NHL are sub-divided into distinct categories based on the differentiation stage of the aberrant B cell and the presence of specific genetic alterations.

Diffuse large B-cell lymphoma (DLBCL) is the most common subtype of B-NHL, accounting for 25–35% of all cases. Although different genetic subsets with distinct genotypic, epigenetic, and clinical characteristics have been recently identified by high-throughput sequencing [1,2], the three molecular subtypes defined in the early 2000s by gene expression profiling, namely germinal center B-cell (GCB)-like, activated B-cell (ABC)-like, and unclassifiable [3], are still widely used in clinics. Patients with ABC-DLBCL or genetic alterations in *MYC* and *BCL2* and/or *BCL6*, called double hit (DHL) or triple hit lymphomas (THL), generally have a poor survival prognosis. The standard of care for DLBCL is an immunochemotherapeutic regimen combining the chemotherapeutic drugs cyclophosphamide, doxorubicin, vincristine, and prednisone (CHOP) with the anti-CD20 monoclonal antibody rituximab (R-CHOP) [4,5].

Follicular lymphoma (FL) is a neoplasm originating from germinal center (GC) cells with a follicular pattern. It is the second most common B-NHL, accounting for 20% of all B-NHL cases. FL is characterized by the t(14;18)(q32;q21) translocation involving the *BCL2* gene, present in 90% of grade 1–2 patients [6,7]. However, the clinical course is mostly indolent; about 20% of patients, despite treatment, relapse or progress to transformed-FL (t-FL), a more aggressive subtype. Treatment usually involves localized radiotherapy for early stages and rituximab combined with chemotherapy regimens like CHOP for advanced stages [7,8].

Burkitt lymphoma (BL) is another GC-derived lymphoma characterized by the deregulation of *MYC* due to translocations such as t(8;14)(q23;q32). Three subtypes have been described, namely endemic, sporadic, and immunodeficiency-associated form, which is mostly found in patients infected with the human immunodeficiency virus (HIV). Although BL is an aggressive neoplasm, most patients respond to intensive chemotherapeutic regimens [9,10].

Marginal zone lymphomas (MZL), accounting for 5–15% of B-NHL, originate from memory B cells. The three described clinical entities are splenic (SMZL), nodal (NMZL), and extra-nodal MZL (EMZL), arising from the marginal zone of the spleen, the lymph nodes, and the mucosa-associated lymphoid tissue (MALT), respectively. EMZL, the most common subtype, is associated with chronic inflammation, such as that derived from *Helicobacter pylori* infections [11,12,13]. Clinical evolution is mostly slow. Treatment usually involves antibiotic treatment for *H. pylori*-positive gastric EMZL, splenectomy for SMZL, radiotherapy for localized disease, and chemotherapy regimens combined with rituximab for advanced stages [14].

Mantle cell lymphoma (MCL) originates from mature B cells in the mantle zone of lymph nodes and accounts for 3–10% of B-NHL. Its molecular hallmark is the t(11;14)(q13;q32) translocation, which leads to the overexpression of cyclin D1 (*CCND1*). MCL has a poor prognosis due to diagnosis often at a disseminated stage and an aggressive clinical evolution. Treatment usually involves immunochemotherapy regimens such as R-CHOP, followed by rituximab maintenance, and autologous stem cell transplantation (ASCT) in fit cases [15,16] (Figure 1).

In conclusion, B-NHL subtypes range in their severity from well-controlled indolent diseases to extremely aggressive forms that have an unmet need for the development of novel therapeutic options. Related B-cell-derived neoplasms further include multiple myeloma (MM) and chronic lymphocytic leukemia (CLL).

### 1.2. Epigenetic Modification of Histone Proteins in B-NHL

Epigenetics was defined in 1942 by Conrad H. Waddington as “the branch of biology which studies the causal interactions between genes and their products, which bring the phenotype into being” [17]. The more scientific knowledge evolved, the more epigenetics became understood as a group of molecular mechanisms constituting a level of memory of previous signals by marking genomic loci and determining the accessibility of embedded genes [18]. Thus, epigenetics is the bridge between the genotype and the phenotype, anchored in the structure and packaging of the genome into chromatin.

The chromatin inside the nucleic compartment is highly compacted and formed by RNA, DNA, and proteins. Importantly, chromatin has a three-dimensional structure that is dynamic and varies, not only between cells of the same or different cell types, but also during the lifespan of a cell itself [19]. Within the interphase nucleus, the chromatin is present as chromosomes, which occupy separate and distinct spaces denominated chromosome territories [20]. Differences in the state of compaction are visible when staining chromatin; less dense regions indicate more loosely packed euchromatin enriched in active transcription, while the strongly stained regions indicate denser heterochromatin harboring repressed genes and tandem repeats such as microsatellites, minisatellites, and transposons [21]. Heterochromatin is frequently enriched at the periphery of the nucleus and on the surface of the nucleolus. Molecular mechanisms exist to transform euchromatin into heterochromatin and vice versa, which allows genes to be expressed differently based on cell type and differentiation state.

The nucleosome is the structural unit of chromatin. The positioning of nucleosomes and their density is the first level of chromatin compaction [22]. In more recent years, attention has been drawn to chromatin motion as a separate phenomenon from compaction status [23]. Nucleosomes consist of 146 pairs of nucleotides wrapped in two loops around an octamer of 8 core histone proteins [24]. More specifically, each nucleosome contains two H2A histones, two H2B histones, two H3 histones, and two H4 histones. Histone H1 is not part of the nucleosome but stabilizes chromatin between nucleosomes to achieve a higher level of structure [25].

Several epigenetic mechanisms operate on the level of the nucleosome. Histone variants can replace replication-coupled histones and provide the nucleosome with different biochemical and biophysical properties [26]. The N-terminal and also some C-terminal tails of histones protrude out of the compact structure of the nucleosome [24] and serve as a platform for many post-translational modifications (PTMs) [27]. PTMs can also occur on the core histone fold, where they directly affect the histone-DNA interaction [28]. Many types of histones PTMs exist that include methylation, acetylation, ubiquitination, SUMOylation, citrullination, glycosylation, ADP-ribosylation, and phosphorylation [27]. The combinatorial nature of PTMs at histone residues led to the controversial hypothesis from Strahl and Allis of a “histone code” on top of the genetic code [29]. Most histone PTMs are catalyzed by enzymes. In the jargon of the chromatin community, we talk about these enzymes as “writers” of PTMs that are recognized by “readers” and removed by “erasers” [27]. The dynamic nature of these mechanisms and their interactions with the transcriptional machinery provide robustness to gene expression programs and a memory of extrinsic or intrinsic stimuli, thereby contributing to the identity and fate of a cell. This precise regulation is perturbed in cancer [30]. The relation with the transcriptional regulation is best understood for the mutual exclusive acetylation and methylation of lysine residues [31].

The writers and erasers of acetylation are histone acetyltransferases (HATs) and histone deacetylase (HDACs), respectively. HATs can be divided into several families: the GNAT family with GCN5; the MYST family including MOZ and Tip60; and the p300/CBP family, among others [32]. Similarly, HDACs are separated into three classes that have zinc-dependent enzymatic activity [33] and the Sirtuins that are NAD^+^-dependent [34]. The transfer of a methyl group from S-adenosyl methionine (SAM) to a lysine or arginine residue is facilitated by histone methyl transferases (HMTs) such as G9a, EZH2, or protein arginine methyltransferases (PRMTs) [35]. The corresponding erasers are histone demethylases (HDMs), of which LSD1 is an important example [35].

Independent from histones, cytosine methylation is a repressive mark occurring on the DNA molecule. DNA methyltransferases (DNMTs) add methyl groups to cytosine bases [36]. Importantly, DNA methylation can only be indirectly removed by TET enzymes and subsequent DNA repair pathways or by dilution through cell divisions [37].

Genes encoding chromatin-modifying enzymes are frequently mutated in B-cell lymphomas. In DLBCL patients, recurrent mutations affect the writers CBP (also named CREBBP), EP300, EZH2, DNMT3A, or KMT2D/MLL4, among others [38]. Chromatin-modifying enzymes can also be strongly overexpressed in B-NHL subtypes, for instance, the arginine methyltransferase PRMT5 in DLBCL and MCL [39]. Furthermore, oncogenic drivers of B-NHL, such as MYC and BCL-6, act as recruitment platforms for chromatin-modifying enzymes resulting in an altered epigenetic landscape. MYC interacts with the acetylation writers p300, GCN5, and Tip60; the erasers HDAC1 and HDAC3; the histone demethylases LSD1 and KDM4B; and the DNA methyltransferase DNMT3A [40]. BCL-6 can recruit CBP, class I, and II HDACs as well as components of the nucleosome remodeling NuRD complex or polycomb proteins [41,42,43,44] (Figure 2).

In conclusion, mechanisms of epigenetic regulation are disrupted in B-NHL through mutations, overexpression, and false recruitment of chromatin-modifying enzymes. By targeting chromatin-modifying enzymes with epidrugs, the field aims at reverting epigenetic changes on chromatin for therapy [45]. The oldest epidrugs are azanucleosides that inhibit DNMT enzymes [46]. In this review, we focus on the current toolkit available to target PTMs on histone residues, providing rationales for the use of epidrugs in B-cell neoplasms and revisiting ongoing clinical trials.

## 2. The Pharmacological Targeting of Histone Acetylation

B-cell lymphoma show reduced levels of histone acetylation compared to untransformed B-cells, which is likely driving the malignant program [47]. The oncogene MYC, for example, silences loci encoding tumor suppressors involving HDAC3 [48]. Thus, targeting HDACs is a promising strategy to re-establish physiological histone acetylation levels in cancer cells and reactivate the expression of tumor suppressor genes. In addition, increasing histone acetylation in B-NHL cells can cause apoptosis induction, cell cycle arrest, DNA damage induction, or reduced proliferation [49]. HDAC inhibition also alters the acetylation status of non-histone substrates like p53, MYC, or NF-κB [50], which can indirectly affect gene expression through altered activity of the transcriptional machinery. Several compounds interfering with histone acetylation have entered the preclinical and clinical stages of drug development. In addition to inhibitors of HDACs, these also include inhibitors of HATs and compounds interfering with readers of the family of bromodomain and extra-terminal domain (BET)-containing proteins (Figure 2). We focus on B-NHL but on occasions extend our discussion to other hematopoietic malignancies.

### 2.1. Extensive Preclinical Development of HDAC Inhibitors

HDAC inhibitors (HDACi) against class I, II, and IV HDACs are grouped based on their chemical properties into hydroxamic acids (TSA, SAHA, panobinostat, rocilinostat), short-chain fatty acids (valproic acid), benzamides (entinostat, mocetinostat), and cyclic tetrapeptides (romidepsin) [49]. Although structurally different, the common mode of action of HDACi against class I, II, and IV lies in chelating the central zinc ion in the catalytic center [51]. On the contrary, sirtuin inhibitors (SIRTi) act as non-competitive inhibitors for NAD^+^, for example, nicotinamide (NAM) [52], or occupy the catalytic center, for example, Ex-527 [53], to prevent enzymatic activity.

HDACi were originally based on the structure of DMSO, which, at certain concentrations, can induce growth arrest in transformed cells [54]. This eventually led to the development of vorinostat, a potent inhibitor of class I and II HDACs [54]. Also, naturally occurring microbial metabolites, such as trichostatin A (TSA), can act as pan-HDACi [55]. In 2015, a database listed a total of 1445 natural or synthetic compounds with HDACi properties [56]. This number is probably surpassed by now as the design and synthesis of HDACi are ongoing by refining side chains of available inhibitors [57]. While most first-generation HDACi showed activity against several classes of HDAC family members (pan-HDACi), novel HDACi are aimed at targeting only one member or class (selective HDACi) to reduce off-target and, eventually, side effects in patients [57]. Novel highly-specific HDACi are, for example, directed against HDAC3 [58] or HDAC6 [59]. However, until now, there has been a huge discrepancy between the number of HDACi on hand and the number of Food and Drug Administration (FDA)-approved HDACi that include only the pan-HDACi vorinostat, romidepsin, panobinostat, and belinostat.

The first requisite before becoming available for clinical trials is to successfully pass preclinical development using a variety of mouse models. Eµ-Myc transgenic mice, for example, recapitulate MYC-induced B-cell lymphomagenesis by restricted *Myc* overexpression in the B-cell compartment using the strong endogenous Eµ enhancer [60]. Phenotypically, Eµ-Myc mice develop immature or mature B-cell lymphomas [60,61] with dramatic tumor heterogeneity between individual mice [62]. However, this should not be interpreted as a weakness of the model, as a recent study utilized Eµ-Myc mice for predicting chemotherapy treatment outcomes of human DLBCL patients [63].

Pan-HDACi such as dacinostat showed strong anti-proliferative effects in panels of B-NHL and MM cell lines [64]. Moreover, apoptosis induction upon pan-HDACi challenge required the pro-apoptotic proteins BID and BIM in primary lymphoma cells from Eµ-Myc mice [65]. Similarly, direct application of vorinostat or panobinostat in vivo in Eµ-Myc mice or lymphoma xenografts extended the median survival significantly via increased apoptosis and upregulated autophagic processes [65,66]. To identify singular HDAC family members mediating pro-tumoral functions, systemic depletions were conducted in Eµ-Myc cells revealing that knock-down of HDAC3 strongly reduced lymphoma cell proliferation and tumor mass in xenografted mice [67]. Besides, only the combinatorial knock-down of *HDAC1* and *HDAC2* led to a huge increase in apoptosis of MYC-driven lymphoma cells [67]. However, in contrast to B-ALL lines, pharmacological targeting of HDAC1/HDAC2 was less effective in reducing the viability of B-NHL cell lines as no upregulation of H2A.X was achieved [64].

HDAC6 has emerged as an interesting target for the development of specific HDAC inhibitors. In *MYC*-induced lymphoma, HDAC6 has a different function than HDAC1 [67]. Indeed, HDAC6 is unique with two catalytic domains, a mainly cytoplasmic location, and substrates such as tubulin and heat-shock proteins [68]. Current research on HDAC6 is fueled by specific inhibitors like rocilinostat, which induces an unfolded protein response (UPR) in DLBCL cells concomitant with overloading of the proteasome [69]. Interestingly, HDAC6 inhibitor treatment of BL and DLBCL cell lines induced MYC degradation, which was accompanied by apoptosis and even prevented lymphomagenesis in Eµ-Myc mice [70]. Of note, HDAC6 seems to have cancer-specific functions as *Hdac6*-knock-out mice showed no obvious phenotype under non-inflammatory conditions [71,72]. This explains the relatively mild side effect profile observed after clinical administration of HDAC6 inhibitors [70], providing an argument for single HDAC member inhibition instead of using pan-HDACi.

Many of the above-made observations for HDACi can be transferred to other lymphoid neoplasms, such as MM [73]. The preclinical model of Vk*MYC mice recapitulates the pathogenesis of MM, including typical signs of disease such as increased antibody production, splenomegaly, osteolytic lesions, and extensively increased numbers of CD138-positive plasma cells [74]. Interestingly, the now FDA-approved HDACi panobinostat prolonged survival of Vk*MYC mice by reducing CD138-positive cells and the M-spike caused by antibody secretion [75]. Besides, rocilinostat, in combination with bortezomib, induced apoptosis in vitro by activation of endoplasmic reticulum (ER) stress in MM cells and significantly increased survival of MM xenografts [76]. A recent study showed that rocilinostat treatment upregulated CD38 surface expression on MM cells which improved targeting by the anti-CD38 monoclonal antibody daratumumab [77]. Rocilinostat also increased CD20 surface expression on B-NHL cells [78], which could be combined with anti-CD20 monoclonal antibody treatment, pointing towards the combination of HDACi with immunotherapy.

The recognition of malignant cells by the immune system is generally known as immune surveillance, and many cancer cells prevent this interaction by blocking immune checkpoints. Importantly, HDAC3 inhibition using the novel molecule BRD3308 upregulated p21 resulting in a cell-intrinsic arrest of proliferation [79]. However, this was only effective in CBP/p300-mutated DLBCL, where BCL6-HDAC3 complexes can be found that repress p21 (*CDKN1A*) transcription [79]. BRD3308 also increased *PDL1* and *HLADR* gene expression in DLBCL, which promoted CD4 and CD8 T-cell recruitment to the lymphoma site in a mouse model [79]. Similar effects on the tumor microenvironment were obtained for HDACi against class I or HDAC6 [80,81].

An important but less studied class of HDACs is class III (Sirtuins). In 2013, Amengual and colleagues combined the general SIRTi NAM with all four FDA-approved pan-HDACi and discovered strong synergies, especially in GC-DLBCL cell lines, by inducing Bcl-6 and p53 acetylation [82]. In addition, λ-MYC transgenic mice—another model for studying the development of mature B-cell lymphomas [83]—were treated with a combination of NAM and romidepsin with synergistic results in survival [82]. A comparison of isoform-specific functions in B-NHL gives a much more detailed view: SIRT1 expression was highly correlated with a worse survival prognosis in DLBCL [84]. In addition, SIRT1 was shown to activate AMPK in primary effusion lymphoma, and SIRT1 inhibition improved the survival of derived xenografts [85]. SIRT3, however, acts as a tumor suppressor in MCL [86] but as an oncogene in DLBCL by its influence on the cancer metabolism [87] and the positive regulation of IDH2 [88]. SIRT6 might also act as an oncogene as its knockdown reduced tumor volume in lymphoma models and increased expression of the negative cell cycle regulator p27 [89]. The clinical use of SIRTi will rely on the development of isoform-specific inhibitors, as currently performed against SIRT2 in B-NHL [90], and will finally shed more light on the impact on specific histone residues and affected target genes.

### 2.2. Preclinical Evaluation of HATs as Targets for B-NHL Therapy

Mutations in genes encoding the HATs, p300 and CBP, can be found in almost 40% of DLBCL and FL, resulting in impaired acetylation of histone residues but also of non-histone proteins, such as BCL-6 and p53 [44]. As p300 and CBP regulate super-enhancer networks in stimulated B-cells, inactivating mutations block terminal differentiation and increase responsiveness towards mitogenic stimuli [91]. Thus, the development of inhibitors against CBP/p300 seems contradictory at first [92]. However, CBP/p300 inhibitors such as C646 were found to reduce *MYC* expression as H3K18ac and H3K27ac marks were absent at the transcriptional start site, and no RNA polymerase II recruitment was present [93]. This, in turn, forced apoptosis induction of lymphoma cells and was associated with a strong tumor reduction in lymphoma xenografts [93].

Evidence for an oncogenic function of the HAT MOZ comes from Eµ-Myc mice where heterozygous knock-out increased median survival almost four-fold [94]. Thus, it is not surprising that MOZ inhibitors also caused cellular senescence and arrested lymphoma growth in murine transplant models [95]. In a similar way, homozygous loss of *GCN5* led to a lifespan extension of Eµ-Myc mice by downregulating the expression of cell cycle-related genes such as *E2f* and *Ccnd1* [96]. This can be pharmacologically explored in BL by drugging GCN5 with the compound MB-3 that induced a G2/M arrest in the cell cycle [97]. Conversely, heterozygous knock-out of the HAT Tip60 was shown to significantly diminish the lifespan of Eµ-Myc mice [98]. Although Tip60 inhibitors targeting the binding site of acetyl-CoA were developed [99], the preclinical data do not favor the use in B-NHL. It should be noted that also HAT activators exist, like YF-2, and are currently explored for the treatment of neurodegenerative diseases. As histone residues are commonly “under”-acetylated in B-NHL, a rationale for the testing of HAT activators is provided.

### 2.3. Experimental Insight in the Utility BET Inhibitors

Bromodomain-containing proteins can recognize acetylated lysine residues and bind to them via the bromodomain. Naturally, proteins harboring bromodomains include chromatin modifiers, such as HATs (CBP/p300), transcription factors (MLL), or transcription-associated proteins (TAF1) [100]. Importantly, the biological effects of inhibiting either the bromodomain or the catalytically active domain of HATs might be very distinct [101]. Among the bromodomain-containing proteins is the subfamily II of special interest. The members of this so-called BET family, i.e., mBRDT, BRD2, BRD3, and BRD4, act as adaptors for transcription factors and contribute to the precise regulation of gene expression [102]. The feasibility of inhibiting bromodomains was first shown for BRD4 [103,104].

BRD4 stimulates the kinase activity of positive transcription elongation factor b (P-TEFb), which in turn promotes Ser2 phosphorylation of RNA polymerase II, and therefore ongoing transcription [105]. Otherwise, RNA polymerase II would pause after transcribing a short strand of DNA, making successful elongation impossible [106]. MYC can recruit P-TEFb at active promoters and enhancers, causing transcriptional amplification [107]. Specifically in MM, BRD4 associates with super-enhancers related to key MM genes, such as *CCDN2*, *PRDM1*, *XBP1*, or *MCL1* [108]. Moreover, BRD4 recruits enzymes involved in histone methylation, including lysine methyltransferases and arginine demethylases, thus, depleting BRD4 reduced H3K36 methylation [109].

BRD4 is druggable by two diazepine-derived compounds; the enantiomer-specific compound (+)-JQ1 and the synthetic histone mimetic I-BET762 (molibresib) [103,104]. (+)-JQ1 accomplished astonishing results in inducing apoptosis, cell cycle arrest, and senescence in BL, DLBCL, and MM cells as well as in derived mouse models [110,111,112]. Mechanistically, (+)-JQ1 blocks BRD4 recruitment by MYC for transcriptional activation and simultaneously directly downregulates expression of *MYC,* resulting in decreased MYC protein levels [110,111,113]. (+)-JQ1 also displaced BRD4 from super-enhancers impacting key oncogenic drivers [108]. Besides, BETi were shown to downregulate the expression of PD-L1 in an MYC-independent manner on lymphoma cells which engaged the immune response in vivo [114].

Subsequently, novel BETi were designed to achieve better suitability for clinical trials by increasing bioavailability or stability and reducing dose-limiting toxicities (DLT). For example, I-BET151 is a novel dimethylisoxazole BETi that is analog to I-BET762 with an enhanced half-life in vivo [115]. I-BET151 induced G1 phase arrest and apoptosis by reducing BCL-2 levels in leukemic cells [116]. Similarly, OTX015 (birabresib) had strong antiproliferative effects in a large panel of B-NHL cell lines from DLBCL, MCL, MZL, and MM subtypes by decreasing expression of E2F3 target genes and leading to a downregulated inflammatory signature, resulting in significant tumor volume reduction in xenografts [116]. The benzoisoxazoloazepine CPI-0610 decreased *MYC* transcripts in vivo and reduced leukemia xenograft tumor growth, which was synergistic with doxorubicin treatment [117]. However, toxicity studies showed that CPI-0610 treatment resulted in lymphoid depletion, hypocellularity of the bone marrow with associated anemia, and thrombocytopenia, among other side effects in animals [117].

Preclinical models also showed that combining BETi with the BH3 mimetic, BCL-2 antagonist venetoclax was beneficial in *MYC*-overexpressing lymphoma cells [118]. Mechanistically, the BCL-2 inhibitor was able to counteract the protective effect of the anti-apoptotic protein, frequently overexpressed in B-NHL, towards BETi-mediated upregulation of the pro-apoptotic BIM and subsequent triggering of apoptosis [118]. This combination significantly reduced the tumor burden of lymphoma xenografts and tremendously increased survival [118]. Combination of (+)-JQ1 and panobinostat synergistically induced apoptosis in MCL cells resistant to the Bruton’s tyrosine kinase (BTK) inhibitor ibrutinib [119]. Also, simultaneous inhibition of BET proteins and blockade of the chemokine receptor CXCR4 potentiated MYC reduction in DLBCL cells and reduced tumor volume of transplanted xenografts [120].

Finally, it should be noticed that non-BET bromodomain inhibitors exist that can be used to target chromatin-modifying enzymes or remodeling complexes. Inhibiting p300 or CBP is possible with bromodomain inhibitors like I-CBP112 or CCS1477 that do not target the catalytic site containing the acetyl transferase activity [121,122]. The bromodomain inhibitor PFI-3 targets SMARCA2, an ATP subunit of the SWI/SNF chromatin remodeling complex, and its association with the HMT NSD2 was found in a specific subset of MM [123]. PFI-3 induced apoptosis in these MM cells, probably by perturbing the expression of myeloma-relevant genes and without effects on BRD4 [123]. BRD9 is another component of the SWI/SNF complex that can be targeted by the non-BETi I-BRD9 [124]. Until today, the Structure Genomics Consortium, a joint venture of industry and academia, has generated high-affinity binders of many human bromodomains by combining different methodologies [125]. The possible usage of such non-BETi should be further explored in specific malignant entities for which rationales exist. Another recent development is dual inhibitors, in which BETi are fused with other inhibitory compounds which generated BET-CBP/p300 inhibitors, BET-BRD7/9 inhibitors, or BET/HDAC inhibitors [126,127,128]. Although strong and consistent anti-tumor activity was already shown for BET-CBP/p300-inhibitors in MM [126], general applications of dual inhibitors in B-NHL remain to be tested.

Taken together, distinct strategies are employed to target the imbalance between HATs and HDACs in the acetylation landscape of lymphoid neoplasms (Table 1). Depending on the disease context, inhibiting readers, writers, and readers of histone acetylation have been shown to be beneficial (Figure 2).

### 2.4. Ongoing Clinical Development of HDACi, BETi and HATi

#### 2.4.1. HDACi

Vorinostat is the most extensively tested HDACi in clinical trials for lymphoma patients, both as a single agent and in combination. Reported overall response rates (ORR) for vorinostat monotherapy in cohorts with different subtypes of B-NHL range from 29% to 40% (NCT00253630, NCT00127140) [129,130]. Specifically, the best response was observed in FL patients, with an ORR of 49%, followed by 27% in MCL and only 5.5% in another trial for DLBCL (NCT00875056, NCT00097929) [131]. Although toxicity was generally manageable, thrombocytopenia was a common adverse event (AE), affecting up to 90% of participants in some of the trials. The combination of vorinostat with rituximab or rituximab-based regimens improved outcomes in some clinical trials. In a cohort of mostly relapsed FL, MCL, and MZL patients, vorinostat and rituximab achieved an ORR of 46% (NCT00720876) [132], increasing up to 65% in a similar cohort treated with rituximab, ifosfamide, carboplatin, and etoposide (R-ICE) (NCT00601718). In MCL, the combination with rituximab and cladribine resulted in an ORR of 39% in relapsed patients and an impressive 97% when used at the frontline (NCT00764517) [133]. In DLBCL, the combination with R-CHOP also achieved remarkable results, with an overall survival (OS) of 86% and progression-free survival (PFS) of 73% at 2 years (NCT00972478). All-grade AEs and serious AEs (SAEs) occurred in more than 70% and 30% of participants, respectively, in the four clinical trials mentioned, being especially relevant the presence of serious febrile neutropenia in 35% of DLBCL participants treated with vorinostat and R-CHOP. Other drug combinations with vorinostat demonstrated promising results as well. When vorinostat was combined with azacitidine, rituximab, and other chemotherapeutic drugs as preconditioning therapy before ASCT, event-free survival (EFS) at 100 days post-transplant was 66% in DLBCL and 100% in FL and MCL. Although some all-grade AEs, such as neutropenia and mucositis, affected over 90% of participants, no SAEs were reported (NCT01983969). The combination with the proteasome inhibitor bortezomib resulted in a modest ORR of 27% in MCL and 8% in DLBCL (NCT00703664), but results were more remarkable as maintenance therapy after ASCT, as B- and T-NHL patients presented 84% OS and 74% EFS at 6.5 years post-ASCT (NCT00992446). The combination of vorinostat, etoposide, and niacinamide was evaluated in a cohort with only four B-NHL patients, but one of them had a complete response (CR) (NCT00691210) [82], while the combination of vorinostat with the aurora A inhibitor alisertib led to 2 CR in a cohort of 12 relapsed and refractory (R/R) DLBCL patients (NCT01567709) [134]. Vorinostat combined with the immunomodulatory drug (IMiD) lenalidomide presented manageable toxicity; however, the trial was terminated early due to low recruitment. The mTOR inhibitor tacrolimus (NCT04220008, NCT03842696), the PARP inhibitor olaparib (NCT03259503), and the PD-1 immune checkpoint inhibitor pembrolizumab (NCT03150329), among others, are also currently evaluated in combination with vorinostat (Table 2).

The rest of HDACi evaluated in B-NHL patients have shown modest results so far. Panobinostat monotherapy, evaluated in a cohort of B- and T-NHL patients, led to median OS and PFS of 15 and 3 months, respectively, and an ORR of 21% (NCT01261247), and similar results were reported in a cohort of DLBCL patients refractory to R-CHOP treatment (NCT01523834). Two separate DLBCL clinical trials (NCT01238692, NCT01282476) reported no improvement of the outcome when combined with rituximab [135]. Panobinostat was also evaluated in combination with the mTOR inhibitor everolimus in 3 clinical trials in a cohort of B-, T-NHL and Hodgkin lymphoma (HL) patients (NCT00918333, NCT00967044, NCT00978432), in which the best response was an ORR of 33%, and OS and PFS of 35 and 4.2 months, respectively. Regarding toxicity, thrombocytopenia was the most notable AE, both in monotherapy and combination trials, reaching a frequency over 90% in some of them.

Romidepsin was evaluated as a single agent in a small cohort of 9 relapsed DLBCL and MCL patients, of which only 1 had a partial response (PR) (NCT00383565). Combinations have been evaluated in lymphoma cohorts formed mainly by patients with T-NHL, who achieved better outcomes than those with B-NHL. The best result was an ORR of 75% in four FL patients treated with romidepsin and pralatrexate (NCT01947140) [136], but much lower ORRs were reported in trials combining romidepsin with azacitidine (NCT01998035) [137] or gemcitabine, dexamethasone, and cisplatin (NCT01846390) [138]. An ongoing clinical trial is testing the combination with lenalidomide (NCT01755975).

Abexinostat monotherapy has shown promising results in FL patients in two clinical trials (NCT00724984, EudraCT-2009-013691-47), leading to an ORR of 56% and a PFS of 10.2 months, but achieving more modest outcomes in MCL and DLBCL [139]. Several ongoing trials are testing abexinostat as a single agent in different subtypes of NHL (NCT03600441, NCT04014696, NCT03936153, NCT03934567) and in combination with ibrutinib (NCT03939182).

Valproic acid has shown potent activity in DLBCL in combination with R-CHOP, achieving OS and PFS at 2 years of 97% and 85%, respectively, and an ORR of 90%. However, toxicity was prominent, as 81% of participants experienced grade 3 or 4 neutropenia, and notable auditory AE was common (NCT01622439) [140].

Fimepinostat monotherapy in DLBCL achieved an ORR of 47%, and median PFS was 3 months, but the combination with rituximab did not improve these outcomes (NCT01742988) [141].

Mocetinostat monotherapy has shown modest activity in DLBCL and FL patients so far, with an ORR below 20% and a PFS below 4 months (NCT00359086) [142].

Belinostat has shown no clinical benefit for lymphoma patients, as no response has been reported either as a monotherapy (NCT00303953, NCT01273155) or in combination with ibritumomab tiuxetan, an yttrium-90-labelled anti-CD20 monoclonal antibody (NCT01686165).

Entinostat (NCT02780804, NCT03179930), quisinostat (NCT00677105), tucidinostat (NCT04661943, NCT04337606, NCT04231448, NCT04025593, NCT04022005, NCT03974243), and rocilinostat (NCT02091063) have been or are currently being evaluated for B-NHL patients as well, either as single agents or in combination, but no results are available at the moment.

#### 2.4.2. HATi

As for HAT inhibitors, only CCS1477-02 has entered clinical development for B-NHL, with an ongoing phase 1 trial for patients with hematological malignancies that is estimated to end in December 2021 (NCT04068597).

#### 2.4.3. BETi

Regarding BET inhibitors, birabresib monotherapy was evaluated in 22 DLBCL patients, which showed an ORR of 10% and a high incidence of AEs, including thrombocytopenia and anemia affecting more than 90% (NCT01713582) [143]. A second trial testing birabresib discontinued enrollment due to the same lack of efficacy (NCT02698189).

RO6870810 as a single agent has demonstrated limited efficacy in DLBCL, with an ORR of 10.5%, median PFS of only 29 days, and prominent toxicity (NCT01987362) [144]. The combination with rituximab and venetoclax has been recently evaluated in lymphoma patients, with no results available yet (NCT03255096).

INCB054329 (NCT02431260) and INCB057643 (NCT02711137) monotherapies were evaluated in cohorts of cancer patients that included 4 and 16 cases of lymphoma, respectively. Both trials were terminated early as all patients discontinued due to lack of responses and notable toxicity [145].

CPI-0610 and FT-1101 have been evaluated in B-NHL patients, both with minimal efficacy: CPI-0610 led to a response in four DLBCL and one FL patient out of 64 cases with B-NHL (NCT01949883), and FT-1101 did not achieve a response in any of the ten B-NHL patients included in the trial (NCT02543879). Lastly, molibresib was recently evaluated as a single agent in a phase 2 trial (NCT01943851), and AZD5153 is being studied in combination with the PARP inhibitor (NCT03205176), but no efficacy results are available yet.

The appearance of DLTs after BETi treatment in clinical studies (Table 2) [146] provides an argument for combining BETi with other compounds to achieve lower effective dosing of BET antagonists. As Basheer and Huntly concluded for BETi, “monotherapy will not suffice to effectively treat complicated hematologic malignancies” [147]. Another argument for combination treatments with BETi is the notion of (+)-JQ1-insensitive genes in B-cell lymphoma likely due to transcriptional rearrangements, which are typical for post-GC lymphomas [148].

## 3. Targeting Histone Methylation and Some Other PTMs

The field of histone methylation has a complex nature due to the large number of players and different configurations of methylation modifications. Lysine residues can be mono-, di-, and tri-methylated, while arginine residues can be mono- or di-methylated, with di-methylation being symmetric or asymmetric [156]. If carefully interpreted, methylation marks are useful for the functional annotation of the genome, in particular, non-coding regulatory regions [157]. Mono-methylation of H3K4 marks active promoter sites and enhancers. Tri-methylation of H3K4 is indicative of a promoter driving the active transcription of its gene, but its co-occurrence with tri-methylation of H3K27 correlates with a poised transcriptional state. Larger regions of tri-methylation of H3K27 alone are indicative of a repressed heterochromatin state (Figure 2).

In DLBCL and FL, a large sequencing study revealed frequent somatic mutations in the PRC2 subunit EZH2 [158]. This is mirrored in the recent re-classification of DLBCL subtypes which now contains a cluster of EZH2^MUT^ lymphomas [1,2]. The frequent tyrosine 641 (Y641) mutation of EZH2 in lymphomas is a gain-of-function mutation that enhances tri-methylation of H3K27 in concert with EZH2^WT^ that performs the H3K27 mono-methylation, eventually shutting down key B-cell genes [157,158]. Moreover, EZH2^MUT^ also reprograms interactions of lymphoma cells with other immune cells in the niche [159]. Besides, mutations in *EZH2* can also be found in BL, and its expression is upregulated in pre-plasmablasts, implicating additional roles in B-cell development and associated pathophysiological conditions [160,161].

### 3.1. Preclinical Development of Histone Methyl Transferase and Demethylase Inhibitors

The overactivity of HMTs in disease provides a rationale for the development of enzymatic inhibitors. Indeed, most drugs against an HMT are available for EZH2, starting from unspecific drugs like DZNep to specific dual EZH1/2 inhibitors, like UNC1999 that is used in B-NHL treatment, and finally, the very selective EZH2i valemetostat and tazemetostat [162]. EZH2 inhibitors specifically target EZH2^MUT^ lymphoma cells, restoring physiological H3K27me3 levels [163,164]. This reactivates expression of genes like *TXNIP* and *TNFRSF21*, resulting in cell cycle arrest, cell death, and reduced tumor load of xenografts [163,164]. However, also non-malignant cells are affected by reduced tri-methylation [164].

The HMT G9a mediates mono- and di-methylation of H3K9, which is followed by tri-methylation via SUV39H1 that is sensitive to chaetocin [165]. Inhibition of SUV39H1 enabled differentiation of leukemic cells and was combined with other epi-drugs [165].

The drug EPZ-5676 (pinomestat) targets specifically the H3K79 methyltransferase DOTL1 [166]. The H3K79 histone residue is located in the core histone fold [28], and inhibition of DOTL1 might be favorable for a specific leukemia subtype [166].

Not only lysine methyltransferases were identified as driving forces of B-NHL but also arginine methyl transferases such as CARM1 and PRMT5. CARM1 inhibition using TP-064 reduced the proliferation of DLBCL cells which was most effective when these cells carried inactivating mutations in genes encoding the HATs CBP or p300 [167]. CARM1 inhibition reduced CBP/p300 target gene expression, as these loci exhibited a loss of H3K4me3 and H3K27ac signals [167]. Besides, treatment with the more bioavailable CARM1 inhibitor EZM2302 reduced tumor volume of p300^MUT^ DLBCL cells in xenograft models, which was highly synergetic with p300/CBP inhibitors [167].

The other example, PRMT5, is overexpressed in some B-cell lymphoma lines in an MYC-dependent manner [168,169]. PRMT5 directly methylates H3R8 and H4R3, which results in hypermethylation and transcriptional repression at target genes, such as *RB1* [168]. Importantly, knock-down of *PRMT5* upregulated pRB protein levels, impaired proliferation of lymphoma cells, and induced cell death as detected by caspase-3 cleavage [39]. PRMT5 also methylates non-histone proteins, which are crucial for ribosome biogenesis [169]. As proliferating lymphoma cells rely on increased translation, knock-out of PRMT5 doubled the lifespan of Eµ-Myc mice [169]. The PRMT5 inhibitor EPZ015666 (GSK3235025) showed striking anti-proliferative properties in vitro and in vivo against MCL, with an IC_50_ in the low nanomolar range [170]. As PRMT5 also directly interacts with BCL-6, it is not surprising that PRMT5 inhibition slowed down the proliferation of DLBCL cells [171].

In addition to the development of HMT inhibitors, efforts are being made to target erasers and readers of histone methylation. A major impact on lymphoma formation has the lysine demethylase LSD1 as it interacts with BCL-6 to repress B-cell termination [172]. Cell type-specific knock-out of LSD1 prolonged survival of lymphoma-prone mice [172], which supports the use of LSD1 inhibitors in B-NHL. The inhibitor GSK-J4 targets the lysine demethylase KDM6B, which is partially overexpressed in B-NHL [173]. Although GSK-J4 did not show strong effects on B-NHL cell proliferation, the combination with other chemotherapeutic agents was highly synergistic in inducing apoptosis [173].

In contrast to acetylation marks that are only read by bromodomain-containing proteins, the diversity of methylation marks is also reflected in a number of different modules able to bind methylated lysine or arginine residues [155]. Interestingly, these methyl-“reader” domains are found in many proteins with different functions. Some examples include chromodomains that can attach to H3K9me2 or H3K9me3, among others, like the heterochromatin factor HP1 [174]. However, the chromodomain of the HAT Tip60 was thought to attach to H3K9me3 or H3K4me1, but this was not confirmed in vitro [175]. The tudor domain found, for example, in JMJD2A or 53BP1, binds methylated histone H3K4 and H4K20 residues [176]. The PHD finger, which is found in several chromatin-modifying enzymes, recognizes H3K4me2 and H3K4me3 but is, surprisingly, also able to bind methylated DNMT enzymes [177,178]. This recalls that there are different ways to inhibit the same enzyme, representing a magnitude of options if appropriate inhibitors are developed. To summarize it in other words: “Only six of the hundreds of known methyl-lysine reader proteins have been targeted” [155].

An interesting example for a bromodomain-containing protein to target is CBX7, a polycomb protein, which is highly expressed in FL [179]. In mouse models, the combination of MYC with CBX7 overexpression resulted in a faster lymphoma development [180]. CBX7 binds via its chromodomain trimethylated H3K9 and H3K27 [180], occupies the *INK4A/ARF* locus, and silences expression of the corresponding tumor suppressors [181]. Recently developed inhibitors against CBX7 [182] might restore the INK4A/ARF-axis in malignant cells and thus, prevent lymphoma cell proliferation.

### 3.2. EZH2 Inhibitors in Clinical Development

Almost the entire clinical development of HMT inhibitors is focused on targeting EZH2. Among the SAM-competitive inhibitors developed to interfere with EZH2, tazemetostat (E7438/EPZ6438) [183] has shown improved potency and pharmacokinetic properties compared to other agents and can be administered orally in animals [184]. This compound is under evaluation in a series of clinical trials in both solid and hematological tumors. In a phase 1 trial (NCT01897571), it showed anticancer activity and a favorable safety profile in patients with R/R B-NHL, as durable objective responses (DOR) were observed in 38% of the cases, including some CRs, with only grade 1 and 2 AEs [185]. In a subsequent open-label, single-arm, multicenter phase 2 trial with *n* = 99 evaluable R/R FL patients receiving 800 mg tazemetostat twice a day until progressive disease or withdrawal, the ORR was 69% in the EZH2^MUT^ cohort (*n* = 45 patients) and 35% in the EZH2^WT^ cohort (*n* = 54 patients). Median DOR was 10.9 months in the EZH2^MUT^ cohort and 13.0 months in the EZH2^WT^ cohort; median PFS was 13.8 months and 11.1 months, respectively. Treatment with tazemetostat was generally well-tolerated, and no treatment-related deaths were observed. Serious treatment-related AEs were reported only in 4% of the 99 patients [186]. Following this study, which confirmed tazemetostat to be a safe and effective new therapeutic drug for patients with R/R FL, the FDA granted accelerated approval for tazemetostat in June 2020. Eligible adult patients either have FL positive for an EZH2 mutation as detected by an FDA-approved test and received at least two prior systemic therapies or have R/R FL with no satisfactory alternative treatment options [187].

In parallel, tazemetostat has also been evaluated in combination studies, firstly in a phase 1b clinical trial with R-CHOP in DLBCL patients. This study determined that a twice-daily administration of 800 mg tazemetostat was the optimum dose for phase 2 evaluation and reported grade 3–4 hematologic AEs in 8/17 patients [152]. Another phase 1b/3 randomized study (NCT04224493) is currently recruiting patients with histologically confirmed FL (grades 1–3A; EZH2^WT^ or EZH2^MUT^ status), treated previously with ≥1 line of chemotherapy, immunotherapy, or chemoimmunotherapy, to determine and assess the recommended phase 3 dose, efficacy, and safety of lenalidomide and rituximab (R^2^) + tazemetostat vs. R^2^ + placebo in patients with R/R FL [153].

GSK2816126 (thereafter GSK126) is another methyltransferase inhibitor capable of inhibiting both EZH2^WT^ and EZH2^MUT^ with similar potency in the low nanomolar range of concentrations and with high selectivity for EZH2 over EZH1 (>100-fold increased potency) and 20 other methyltransferases [187]. This compound has been shown to display significant anti-tumor activity in xenograft models of EZH2^MUT^ DLBCL [188]. Based on these preclinical results, a multicenter, open-label, dose-escalation study (NCT02082977) was carried out in patients with advanced cancers, including 20 B-NHL cases, to evaluate the safety and to determine the maximum-tolerated dose (MTD), the pharmacokinetics, and pharmacodynamics of GSK126. All patients had at least one AE, with the most common being nausea and vomiting in about half of the patients. The results showed insufficient evidence of clinical activity, with only one patient showing a partial radiologic response and 34% of patients having achieved an SD while 51% of them underwent disease progression [188]. Therefore, the compound was not further investigated in clinical settings. One explanation for these discrepancies between preclinical and clinical studies was the presence of myeloid-derived suppressor cells, which may be increased in immunocompetent hosts after treatment, thus impairing the antitumoral activity of the compound [189].

EI1 is a cell-permeable indolocarboxamide compound that acts as a selective SAM-competitive inhibitor of the EZH2 methyltransferase activity with an IC_50_ of 13–15 nM against H3K27me marks, showing activity against EZH2^WT^ or EZH2^MUT^-containing PRC2. This compound has been shown to suppress cellular H3K27 di-methylation and tri-methylation in the low micromolar ranges of concentrations in DLBCL cultures and to exert anti-proliferative activity in DLBCL cell lines with either EZH2^WT^ or EZH2^MUT^ [190]. EPZ011989 is another selective and orally bioavailable EZH2 inhibitor that equipotently inhibits EZH2^WT^ or EZH2^MUT^ with an inhibition constant (Ki) of <3 nM and with the capacity to significantly inhibit DLBCL tumor growth in mouse xenografts [191].

CPI-169, an indole-based selective EZH2 inhibitor, first showed significant antitumor activity and pharmacodynamic (PD) target engagement in a DLBCL mouse xenograft model of EZH2^MUT^, but with limited oral bioavailability [192]. Nonetheless, this compound was able to exert significant antitumor activity in this model when administered intraperitoneally, together with a synergistic tumor growth inhibition when combined with the BRD4 inhibitor, CPI-203 [193]. CPI-1205 (lirametostat), an orally available, subsequently developed compound by the same company, harbored IC_50_ values of 2 nM and 52 nM for EZH2 and EZH1, respectively. This compound exerted antitumoral activity in in vitro and in vivo models of EZH2^MUT^ DLBCL by selectively binding to the EZH2 catalytic pocket [194]. A phase 1 sequential dose-escalation and expansion study (NCT02395601) was launched in patients with progressive B-cell lymphomas to define MTD and DLT of CPI-1205. Preliminary results from this trial revealed that most treatment-related AES were grade 2 or lower, with no DLTs reached. Authors reported one CR, five SD (three remained on SD  ≥ 6 months) with target engagement confirmed by immunohistochemical detection of H3K27me3, and concluded that the drug was well tolerated with manageable toxicities and evidence of antitumor activity [154]. CPI-0209 is a second-generation EZH2 inhibitor that showed robust PD effects in a recent phase 1 trial in different subtypes of solid tumors, making this agent well-tolerated with manageable treatment-related AEs [195].

Activated IGF-1R, PI3K, and MAPK pathways confer resistance to EZH2 inhibition in DLBCL [196]. To improve antitumor efficacy of targeting methyltransferase activity, simultaneous targeting of both EZH2 and its homolog EZH1 [197] has been considered. UNC1999 was the first orally bioavailable SAM-competitive inhibitor of both EZH2^WT^ and EZH2^MUT^ and EZH1, having the ability to selectively kill EZH2^MUT^ DLBCL cell lines [198]. More recently, two orally bioavailable EZH1/2 dual inhibitors, (R)-OR-S1 and (R)-OR-S2, demonstrated improved antitumor activity in in vitro and in vivo models of EZH2^MUT^ DLBCL with no significant toxicity [199]. An ongoing phase 1 study (NCT02732275) is showing that the clinical lead, DS-3201b, exerts early clinical activity and has the potential to be an orally available, novel therapeutic option for B-NHL [200]. Among the latest molecules, the lactam-derived and orally bioavailable EZH2 antagonist PF-06821497 [201] and SHR2554 are potent and orally available EZH2 antagonists, being evaluated in open-label, multi center, phase 1 dose-escalation trials with R/R FL and DLBCL patients (NCT03460977) and R/R mature lymphoid neoplasms (NCT03603951), respectively. Apart from EZH2 inhibitors, the PRMT5 inhibitor EPZ015938 (GSK3326595) with improved activity, when compared to EPZ015666, is currently being used in a dose-escalation study (NCT02783300) with B-NHL patients.

EZH2 has become a hotspot of research in the field of GC-derived B-cell lymphoma, and this interest rapidly led to the emergence of several new drugs that made a quick transfer from bench-to-bedside. These advances also led to the discovery of more functions and roles of EZH2 in tumorigenesis. However, tazemetostat is the only molecule approved to date to treat a subgroup of patients. Continuous efforts are still to be made in the development of (1) EZH2 inhibitors with higher efficiency, lower toxicity, and higher selectivity and (2) dual EZH1/EZH2 antagonists able to counteract EZH1 compensatory effects in EZH2-compromised cells.

### 3.3. Targeting Other Posttranslational Modifications of Histones

The distinct patterns of epigenetic aberrations may provide novel therapeutic approaches targeting histone modifications. Besides histone methylation and acetylation, the N-termini of histone tails are modified by many other processes, such as phosphorylation, glycosylation, ubiquitination, and sumoylation [202,203,204,205]. However, these modifications are much less explored in the B-NHL field. Here, we discuss the first observations suggesting that interfering with histone phosphorylation, ubiquitination and sumoylation could have therapeutic potential.

Histone H3 has a crucial role in chromatin compaction and chromosome formation [206]. Phosphorylation of H3 at serine 10 (phospho-H3) results in immediate condensation of the chromatin in the late G2 phase of the cell cycle, and thus, is a marker present in dividing cells [207]. Indeed, mitotic activity determined by phospho-H3 counting has been experimentally and clinically used to measure cell proliferation in different types of cancer, including lymphoma [207,208,209,210]. Thus, the discovery of compounds that target histone phosphorylation is an interesting approach for the treatment of oncological patients. In their study, Méhes et al. [211] showed an enhanced G2 phase transition due to the abrupt chromatin phosphorylation in neoplastic B cells. Moreover, during the G2 phase, phospho-H3 co-localizes with the Aurora B kinase, which is known to be frequently overrepresented in NHL patients [211,212]. Interestingly, both preclinical [213] and clinical [214] data have supported the potential for the Aurora kinase inhibition as a new therapy for DLBCL patients. Similarly, small molecule inhibitors of another enzyme that phosphorylates histone H3, Haspin, have been developed as efficient anti-mitotic cancer therapeutics [215]. In B-cell lymphoma xenograft models, the oral administration of the Haspin inhibitor SEL120 revealed encouraging results towards this potential treatment [216].

Several novel therapeutic approaches are being evaluated for the management of R/R B-cell lymphomas [217]. It has been shown that Janus kinase (JAK) signaling is deregulated in several hematologic malignancies [218]. Interestingly, Rui et al. [219] demonstrated that JAK1 contributes to tumor growth and survival of ABC-DLBCL cells through a regulatory mechanism dependent on phosphorylation of histone H3 at tyrosine 41, which supports the development of JAK1 inhibitors for ABC-DLBCL therapy. However, some contradictory data showed an increased risk of B-cell lymphoma development in patients treated with JAK inhibitors by the impairment of immune surveillance [220,221,222], highlighting the importance of assessing the clinical impact of these drugs. Although an increase in phospho-H3 has been associated with poor prognosis in several cancers, the exact mechanisms and the clinical relevance of the histone H3 phosphorylation pattern in B-cell lymphoma requires further clarification.

Recent studies have suggested that sumoylation, a PTM in which a small ubiquitin-like modifier (SUMO) is covalently conjugated to lysine residues, is involved in the regulation of several cellular key processes, such as chromatin occupancy, and is most often associated with reduced gene expression [223]. It has been shown that TRAF6, an E3 ubiquitin ligase correlated with poor prognosis of DLBCL patients [224], can be modified by sumoylation, which in turn may repress the gene transcription through recruitment of HDACs in B cells [225]. Although the role of TRAF6 regulation in lymphomagenesis is still unclear, its activity may have therapeutic implications in DLBCL.

Histone ubiquitination is another important PTM, as its deregulation drives oncogenesis by altering the transcription of key tumor suppressors and oncogenes, which may promote cell proliferation [226]. Indeed, genomic data showed that the genes encoding histone E3 ubiquitin ligases and deubiquitinating enzymes are frequently altered in cancers. The pharmacological inhibition of BMI1 (an activator of the RING1 E3 ubiquitin ligase) and H2AK119ub1 levels both induced apoptosis and prolonged survival in xenograft models of acute myeloid leukemia [227]. Moreover, histone H2B has been shown to be overexpressed and strongly ubiquitinated at K120, which is related to NKX2-1 expression and chromosomal rearrangements in DLBCL cells [228]. In this study, the authors found an aberrant activity of NKX2-1 by in silico expression analysis performed in 204 DLBCL patient samples and suggested that the permissive chromatin structure at the *NKX2-1* gene may be associated with deregulation of histone ubiquitination enzymes in DLBCL. However, the therapeutic potentials of NKX2-1 through histone modification require further examinations.

In conclusion, further insights are needed before the development of epigenetic drugs targeting other histone modifications than acetylation and methylation would be warranted for B-NHL and especially its aggressive subtypes.

## 4. Conclusions and Future Directions

Epigenetic alterations have been shown to play a significant role in a wide range of lymphoid and myeloid neoplasms, mainly in regard to their impact on the regulation of gene expression. In the last 10 to 20 years, major advances have been made in the comprehension of the role of histone modifiers in normal and malignant hematopoiesis. This has spurred the simultaneous development of hundreds of small molecule inhibitors able to target dysregulated or mutated histone modifiers. Despite the remarkable preclinical activity of some of these molecules in selected cancer subtypes, and especially in the relapse setting, these new epigenetic therapies are generally associated with modest overall response rates. Thus, additional in vitro/in vivo studies are still required to develop more effective, more selective, and less toxic epigenetic drugs.

To bypass the current limitations in epidrug development, the field has started to implement high-throughput molecular tools for the identification of disease-specific epigenetic alterations. In parallel, genetic screenings have been demonstrated to be powerful tools able to identify novel drug targets for combination therapies. In particular, strategies aiming at synthetic lethality with disease-specific mutations and other drugs are expected to limit the off-target effects of monotherapies. Another interesting approach is the development of dual inhibitors simultaneously targeting unrelated classes of chromatin-modifying enzymes. Rational-based treatment choices, in particular for combination therapies, will require the development of response-predicting biomarkers. The use of epigenetic signatures has great potential to provide these biomarkers. In any case, future studies will need to consider cancer therapies in the context of their tumor microenvironment, including immune effector cells. The development of ex vivo organoid-like models for B-NHL is needed to accelerate preclinical drug testing.

## Figures and Tables

**Figure 1 ijms-23-00253-f001:**
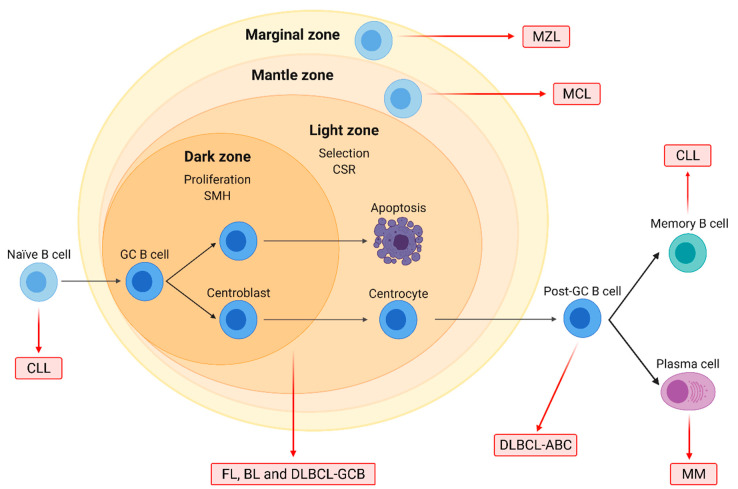
Origin of the major B-cell non-Hodgkin lymphoma (B-NHL) subtypes. Naïve B-cells form germinal centers (GC) after interacting with antigens. In the dark zone, centroblasts proliferate and undergo somatic hypermutation (SMH), while in the light zone, centrocytes are selected based on BCR affinity and undergo class-switch recombination (CSR). GC cells are the normal counterparts of follicular lymphoma (FL), Burkitt lymphoma (BL), and diffuse large B-cell lymphoma (DLBCL) of the GC subtype (GCB). DLBCL of the activated B-cell (ABC) subtype originates from post-GC cells, and multiple myeloma (MM) arises from differentiated plasma cells. Chronic lymphocytic leukemia (CLL) may originate from either naïve or differentiated memory B cells. Mantle cell (MCL) and marginal zone lymphoma (MZL) arise from B cells located on the mantle and the marginal zone of lymphoid follicles, respectively.

**Figure 2 ijms-23-00253-f002:**
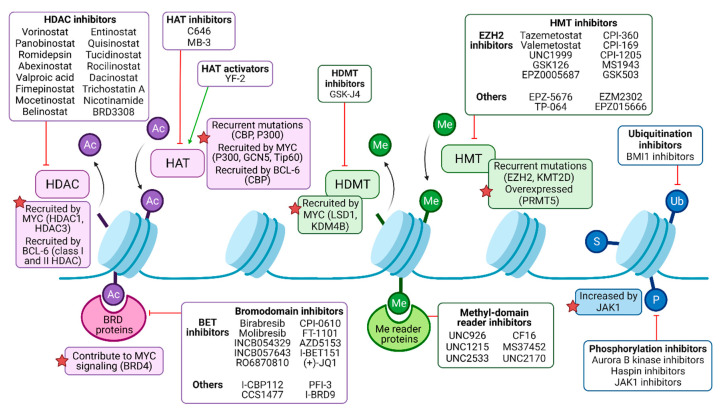
Pharmacological modulation of deregulated histone modifiers in B-NHL. Histone acetylation (Ac) is catalyzed by histone acetyltransferases (HATs), frequently recruited by oncogenic drivers MYC and/or BCL-6 in malignant B cells, and may be targeted by either activators or inhibitors. Histone deacetylases (HDACs) mediate deacetylation and are the target of numerous inhibitory drugs, as some of them present recurrent mutations and B-NHL and may be recruited by MYC and/or BCL-6 as well. Bromodomain (BRD)-containing proteins can bind to acetylated residues, enhancing oncogenic signaling (such as MYC program, in the case of BRD4), and can be targeted by pan or isoform-specific inhibitors. Methylation (Me) is regulated by histone methyltransferases (HMTs) and histone demethylases (HDMTs). HMTs are common targets of epigenetic drugs, especially the EZH2 subunit of the polycomb repressor complex 2 (PRC2), since it is recurrently mutated in B-NHL. HDMT inhibitors can partially counteract MYC signaling, among other effects. Me domain reader proteins, such as those containing chromodomains, may be targeted by inhibitors as well. Other histone modifications include ubiquitination (Ub), sumoylation (S), and phosphorylation (P), but their role in B-NHL pathogenesis and their targeting require further studies.

**Table 1 ijms-23-00253-t001:** Drugs approved as second-line therapy for relapsed/refractory B-NHL and other lymphomas.

Drug		Type	Indication	Year of Approval
Vorinostat	Zolinza	HDACi	CTCL	2006
Romidepsin	Istodax	HDACi	CTCL	2009
			PTCL	2011
Belinostat	Beleodaq	HDACi	PTCL	2014
Panobinostat	Farydak	HDACi	MM	2015
Tazemetostat	Tazverik	EZH2i	R/R FL with or without mEZH2	2020
			Sarcoma	2020

Abbreviations: CTCL, cutaneous T-cell lymphoma; FL, follicular lymphoma; HDAC, histone deacetylase; MM, multiple myeloma; PTCL, peripheral T-cell lymphoma; R/R, relapsed and refractory.

**Table 2 ijms-23-00253-t002:** Current clinical trials evaluating epi-drugs as single agents and/or in combination regimens in B-NHL patients.

Drug/Regimen	Type	Trial ID	Phase	Number of Patients	Disease	Response	Toxicity	Ref
Vorinostat	HDACi	NCT00253630	2	35	FL, MCL, MZL	ORR = 29% OS (2 years) = 77% PFS (2 years) = 37%	26% SAEs: TP (8.5%) 100% AEs: TP (86%)	[130]
Vorinostat	HDACi	NCT00127140	1	10	FL, MCL, DLBCL	ORR = 75%/50%/0%	Grade 3–4 AEs: NP (30%) AEs: TP (70%), anemia (70%), leukopenia (60%)	[129]
Vorinostat	HDACi	NCT00875056	2	56	FL, MCL	ORR = 49%/28%	23%/27% SAEs: TP (5/9%) 100% AEs: TP (95/91%), diarrhea (72%/82%), NP (72%/64%)	[149]
Vorinostat	HDACi	NCT00097929	2	18	DLBCL	ORR = 5%	39% SAEs AEs: diarrhea (61%), fatigue (50%), nausea (39%)	[131]
Vorinostat +rituximab	HDACi	NCT00720876	2	28	FL, MCL, MZL, LPL	ORR = 46% PFS = 29 months	43% SAEs: thrombosis (13%) 100% AEs: fatigue (87%), diarrhea (80%), nausea (73%)	[132]
Vorinostat +R-ICE	HDACi	NCT00601718	1/2	29	DLBCL, MCL, MZL	ORR = 66%	35% SAEs: NP (10%) 86% AEs: hypophosphatemia (41%), hypokalemia (34%)	[149]
Vorinostat +rituximab +cladribine	HDACi	NCT00764517	2	49	MCL (39 frontline/10 R/R)	ORR = 97%/30% OS = 25/6 months PFS = 20/15 months	46%/50% SAEs: NP (23%/22%) 72%/78% AEs: TP (36%/28%), NP (23%/28%)	[133]
Vorinostat + R-CHOP	HDACi	NCT00972478	1/2	83	DLBCL	ORR = 81% OS (2 years) = 86% PFS (2 years) = 73%	68% SAEs: NP (35%), anemia (22%) 100% AEs: anemia (85%), TP (59%), fatigue (74%)	[149]
Vorinostat +azacitidine +rituximab and others (pre-ASCT)	HDACi	NCT01983969	1/2	26	DLBCL	EFS (100 days post-transplant) = 65%	0% SAEs 100% AEs: NP (97%), mucositis (93%), nausea (90%)	[149]
Vorinostat +bortezomib	HDACi	NCT00703664	2	65	MCL. DLBCL	ORR (9 years) = 27%/8% PFS = 7.6 months/1.8 months	38%/56% SAEs: TP (0%/15%) 100% AEs: TP (81%/67%), diarrhea (85%/62%)	[149]
Vorinostat +bortezomib (post-ASCT)	HDACi	NCT00992446	2	19	DLBCL, FL, MCL, T-NHL	OS (6.6 years post ASCT) = 84% EFS (6.6 years post ASCT) = 74%	33% SAEs: all <10% 100% AEs: NP (68%), TP (10%)	[149]
Vorinostat +niacinamide +etoposide	HDACi	NCT00691210	1	25	DLBCL, FL, HL	ORR = 24%	Grade 3–4 AEs: TP (12%), infection (12%) AEs: fatigue (84%), nausea (80%), diarrhea (72%)	[82]
Vorinostat +alisertib	HDACi	NCT01567709	1	12	DLBCL	ORR = 17%	Grade 3–4 AEs: NP (22%), leukopenia (18%), anemia (17%)	[134]
Panobinostat	HDACi	NCT01261247	2	39	DLBCL, MZL, BL	ORR = 21% OS = 14.9 months PFS = 3.1 months	83% SAEs: TP (80%), NP (29%) 93% AEs: fatigue (85%), diarrhea (76%), nausea (72%)	[149]
Panobinostat	HDACi	NCT01523834	2	35	DLBCL (R/R)	ORR = 17% OS = 7.6 months PFS = 2.4 months	35% SAEs: all <10% 23% AEs	[149]
Panobinostat +rituximab	HDACi	NCT01238692	2	40	DLBCL (21 single/19 combo)	ORR = 29%/26%	Grade 3–4 AEs: TP (71%/68%), NP (24%/32%) AEs: TP (76%/79%), diarrhea (76%/58%), nausea (71%/58%)	[135]
Panobinostat +rituximab	HDACi	NCT01282476	2	18	DLBCL	ORR = 11% PFS (6 months) = 6%	56% SAEs: TP (33%) 100% AEs: fatigue (72%), anemia (67%), TP (67%)	[149]
Panobinostat +everolimus	HDACi	NCT00918333	1/2	116	DLBCL, FL, BL, MZL, HL, T-NHL	ORR = 33% OS = 35 months PFS = 4.2 months	24% SAEs: all <10% 99% AEs: decreased Hb (99%), TP (91%), fatigue (90%)	[149]
Panobinostat +everolimus	HDACi	NCT00967044	1/2	30	NHL, HL	N/A	64% SAEs: TP (63%), NP (47%) 100% AEs: fatigue (83%), hyperglycemia (63%), mucositis (60%)	[149]
Panobinostat +everolimus	HDACi	NCT00978432	2	33	DLBCL	ORR = 15%	25% SAEs: all <10% 100% AEs: TP (73%), diarrhea (58%), fatigue (48%)	[149]
Romidepsin	HDACi	NCT00383565	2	9	DLBCL, MCL	ORR = 11% OS = 20 months PFS = 4 months	67% SAEs: TP (22%) 100% AEs: TP (89%), anemia (79%), lymphopenia (67%)	[149]
Romidepsin +pralatrexate	HDACi	NCT01947140	1/2	7	FL, DLBCL, BL	ORR = 75% (FL) OS = 34 months PFS = 1.8 months	Grade 3–4 AEs: anemia (29%), TP (28%), NP (14%) Grade 1–2 AEs: nausea (66%), fatigue (52%), anorexia (24%)	[136]
Romidepsin +azacitidine	HDACi	NCT01998035	1/2	20	DLBCL, FL, HL	ORR = 10% PFS = 2.5 months	Grade 3–4 AEs: NP (42%), lymphopenia (42%), TP (27%), 100% grade 1–2 AEs: hyperglycemia (81%), nausea (54%), vomiting (46%)	[137]
Romidepsin +GDP	HDACi	NCT01846390	1	20	DLBCL, PTCL	ORR = 50% OS = 5.5 months (DLBCL) PFS = 2 months (DLBCL)	60% SAEs: 1 grade 5 sepsis Grade 2–4 AEs: infection (75%), TP (55%), NP (30%), anemia (30%)	[138]
Abexinostat	HDACi	NCT00724984	1/2	30	FL, MCL	ORR = 56% FL/21% MCL	38%/36% SAEs: all <10% 100%/93% AEs: TP (63%/29%), nausea (69%/50%), diarrhea (50%/50%)	[149]
Abexinostat	HDACi	EudraCT-2009-013691-47	2	100	FL, DLBCL, MCL, MZL, T-NHL, CLL	ORR = 56% FL/31% DLBCL PFS = 10.2 months FL/2.8 months DLBCL	73% SAEs: TP (54%), NP (11%) 82% grade 3–4 AEs: TP (80%), NP (27%), anemia (12%) 98% AEs, any grade	[139]
Valproic acid +R-CHOP	HDACi	NCT01622439	1/2	33	DLBCL	ORR = 90% OS (2 years) = 97% PFS (2 years) = 85%	Grade 3–4 AEs: NP (81%), TP (33%), infection (27%) Auditory AEs	[140]
Fimepinostat +rituximab	HDACi	NCT01742988	1	37	DLBCL (25 single/12 combo)	ORR = 47%/18% PFS = 5.7/1.3 months	28% SAEs 43% grade 3–5 AEs: TP (32%), NP (16%) AEs: diarrhea (57%), TP (54%), fatigue (41%)	[141]
Mocetinostat	HDACi	NCT00359086	2	72	NHL (41 DLBCL/31 FL)	ORR = 19%/12% OS = 12 months/N.R. PFS = 2 months/3.7 months	36% SAEs: all <10% 57% grade 3–4 AEs: fatigue (24%), NP (15%), TP (12%) 99% AEs: fatigue (75%), nausea (70%), diarrhea (61%)	[142]
Belinostat	HDACi	NCT00303953	2	22	DLBCL, BL, PMBCL	ORR = 0% OS = 0.9 years PFS = 0.2 years	15% SAEs: all <10% 90% AEs: fatigue (40%), nausea (40%)	[149]
Belinostat +rituximab +ibritumomab tiuxetan	HDACi	NCT01686165	2	5	DLBCL	ORR = 0% PFS (2 years) = 0%	20% SAEs: thrombosis (20%) 60% grade 3–4 AEs: TP (40%), pain (20%), hypoglycemia (20%) 100% AEs: nausea (80%), pain (60%), TP (60%)	[150]
Birabresib	BETi	NCT01713582	1	33	DLBCL, FL, MCL, BL, MZL	ORR = 10%	AEs: TP (96%), anemia (91%), neutropenia (51%)	[143]
Birabresib	BETi	NCT02698189	1	6	DLBCL	ORR = 17%	17% SAEs: infection (17%) 100% AEs: TP (50%), abdominal pain (50%), diarrhea (50%)	[149]
RO6870810	BETi	NCT01987362	1	19	DLBCL	ORR = 10.5% PFS = 29 days	53% SAEs 74% grade 3–4 TRAEs: all <10% 95% all-grade TRAEs: fatigue (42%), nausea (31%), diarrhea (26%)	[144]
INCB054329	BETi	NCT02431260	1/2	4	Lymphoma	ORR = 0%	23% grade 3–4 TRAEs: TP (13%) 78% all-grade TRAEs: nausea (35%), TP (33%), fatigue (29%)	[145]
INCB057643	BETi	NCT02711137	1/2	16	Lymphoma	ORR = 25% (FL)	36% grade 3–4 TRAEs: TP (18%), anemia (10%) 86% all-grade TRAEs: TP (32%), fatigue (30%), nausea (30%)	[145]
CPI-0610	BETi	NCT01949883	1	64	DLBCL, FL	ORR = 7%	TRAEs: TP (45%), fatigue (34%), nausea (27%)	[151]
FT-1101	BETi	NCT02543879	1	10	NHL	ORR = 0%	Grade 3–4 TRAEs: pleural effusion (20%) TRAEs: diarrhea (60%), nausea (40%), pleural effusion (40%)	[152]
Tazemetostat +R-CHOP	EZH2i	NCT02889523	1b	17	DLBCL	mCR = 76%	Grade 3–4 AEs: NP (47%), leukopenia (29%), constipation (24%) AEs: constipation (59%), nausea (59%), vomiting (53%), NP (47%)	[153]
Tazemetostat +lenalidomide +rituximab	EZHi	NCT04224493	1b/3	518 (planned)	FL	N/A	N/A	[154]
Lirametostat	EZHi	NCT02395601	1	32	DLBCL, FL, MZL	ORR = 3% SD = 15%	Grade 3–4 TRAEs: lymphopenia (9%), nausea (3%), anemia (3%) TRAEs: nausea, diarrhea, anemia, fatigue (all >5%)	[155]

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
