# Peer review of "Histone Modifications and Their Targeting in Lymphoid Malignancies"

_ijms, 2021, doi:10.3390/ijms23010253_

Round 1
Reviewer 1 Report
Fernandez-Serrano et al. provide here an extensive review of alterations affecting histone modifications in lymphoid malignancies and how they can be therapeutically targeted. They cover both functional and efficacy aspects derived from pre-clinical studies, as well as the clinical evaluation of numerous small molecules inhibitors developed. The molecular mechanisms involved in these alterations in B-Cell malignancies and their impact of cell signaling are quite exhaustively described. The sections covering clinical developments, very detailed, seem to me up to date and provide a new perspective on the clinical potential of these new molecules in different lymphoma subtypes.This paper is dense, but very well documented and written, and therefore pleasant to read. However, some sections (such as 2.4) could be divided in several sub-sections (HDACi, BETi and HATi) to improve readability.
Overall, I would definitely recommend the publication of this review article in IJMS.
Author Response
Fernandez-Serrano et al. provide here an extensive review of alterations affecting histone modifications in lymphoid malignancies and how they can be therapeutically targeted. They cover both functional and efficacy aspects derived from pre-clinical studies, as well as the clinical evaluation of numerous small molecules inhibitors developed. The molecular mechanisms involved in these alterations in B-Cell malignancies and their impact of cell signaling are quite exhaustively described. The sections covering clinical developments, very detailed, seem to me up to date and provide a new perspective on the clinical potential of these new molecules in different lymphoma subtypes. This paper is dense, but very well documented and written, and therefore pleasant to read.
Comment 1) However, some sections (such as 2.4) could be divided in several sub-sections (HDACi, BETi and HATi) to improve readability.
Answer: we sincerely acknowledge the kind evaluation of the reviewer. On her/his request, we have subdivided the section 2.4 in 3 new subscetions: 2.4.1. HDACi, 2.4.2. HATi and 2.4.3 BETi.
Reviewer 2 Report
Fernández-Serrano et al. review describes the current knowledge about the function of histone modifications and their relative contribution to the pathogenesis of the lymphoid malignancies. In particular, the authors focused on B-cell non-Hodgkin lymphoma (B-NHL) and the potential therapeutic strategies. The present review includes a complete and critical view of this topic. Although the manuscript is very well written and organized, this reviewer believes that the following minor concerns need to be addressed:
- In Figure 2 the authors show the drugs commonly used to target the main epigenetic alterations. They should add the information about those drugs, which are specifically used for the lymphoid malignancies and also in particular for B-NHL. Alternatively the authors could insert a new Figure, Figure 3 graphically describing the inhibitors and their associated histone modifications mainly involved in the pathogenesis of B-NHL.
- In Table 2 the authors should clarify the information about the N, which I believe is the number of patients.
Author Response
Fernández-Serrano et al. review describes the current knowledge about the function of histone modifications and their relative contribution to the pathogenesis of the lymphoid malignancies. In particular, the authors focused on B-cell non-Hodgkin lymphoma (B-NHL) and the potential therapeutic strategies. The present review includes a complete and critical view of this topic. Although the manuscript is very well written and organized, this reviewer believes that the following minor concerns need to be addressed:
Comment 1) In Figure 2 the authors show the drugs commonly used to target the main epigenetic alterations. They should add the information about those drugs, which are specifically used for the lymphoid malignancies and also in particular for B-NHL. Alternatively the authors could insert a new Figure, Figure 3 graphically describing the inhibitors and their associated histone modifications mainly involved in the pathogenesis of B-NHL.
Answer: we would like to point out that all the drugs displayed in the original Figure 2 have been or are currently being used for the preclinical/clinical treatment of epigenetic alterations in B-NHL, as discussed in details in chapters 2 and 3. On the request of the reviewer, we have indicated in the new figure 2 below, the factors as well as the specific alterations being specifically modulated by these different agents in B-NHL.
Figure 2. Pharmacological modulation of deregulated histone modifiers in B-NHL. Histone acetylation (Ac) is catalyzed by histone acetyltransferases (HATs), frequently recruited by oncogenic drivers MYC and/or BCL-6 in malignant B cells, and may be targeted by either activators or inhibitors. Histone deacetylases (HDACs) mediate deacetylation and are the target of numerous inhibitory drugs, as some of them present recurrent mutations and B-NHL and may be recruited by MYC and/or BCL-6 as well. Bromodomain (BRD)-containing proteins can bind to acetylated residues, enhancing oncogenic signaling (such as MYC program, in the case of BRD4), and can be targeted by pan or isoform-specific inhibitors. Methylation (Me) is regulated by histone methyltransferases (HMTs) and histone demethylases (HDMTs). HMTs are common targets of epigenetic drugs, especially the EZH2 subunit of the polycomb repressor complex 2 (PRC2), since it is recurrently mutated in B-NHL. HDMT inhibitors can partially counteract MYC signaling, among other effects. Me domain reader proteins, such as those containing chromodomains, may be targeted by inhibitors as well. Other histone modifications include ubiquitination (Ub), sumoylation (S), and phosphorylation (P), but their role in B-NHL pathogenesis and their targeting require further studies.
Comment 2) In Table 2 the authors should clarify the information about the N, which I believe is the number of patients.
Answer: We are grateful to the reviewer for her/his comment and have changed the headline of the column for “Number of patients”.